## RESEARCH ARTICLE

# Adaptive evolution of cytochrome b in songbirds

Hagai Rottenberg*

## ABSTRACT

The mitochondrial bc1 complex catalyzes the oxidation of ubiquinol by reducing cytochrome c. Cytochrome b, the catalytic core of bc1, generates superoxide during the oxidation of ubiquinol. Excessive superoxide production is known to accelerate aging and neurodegeneration. Songbirds (oscine, Passeri) exhibit lower production of mitochondrial reactive oxygen species (ROS) and greatly accelerated evolution of cytochrome b, relative to all other modern birds, suggesting adaptive selection for lower generation of ROS. Here, we identified songbird-specific substitutions in modern bird's cytochrome b amino-acid sequences and examined the high-resolution structures of the chicken bc1 complex in an effort to predict the effect of these substitutions on the function of bc1. Many of the songbird-specific substitutions cluster around sites that are critical for the function of bc1. One cluster of substitutions interacts with heme B$\mathbf{H}$. A second cluster of substitutions interacts with residues in the ubiquinone reduction site, Qi. Both groups of substitution may affect the rate of reduction of ubiquinone at the Qi site. Another cluster of cytochrome b substitutions interacts with the hinge region of the Rieske protein that transfers electron from cytochrome b to cytochrome c1. These songbird-specific substitutions appear to be selected to modulate the rate of both ubiquinol oxidation at the Qo site and ubiquinone reduction at the Qi site thereby modulating the rate of superoxide production. These findings are compatible with the hypothesis that cytochrome b evolution in songbirds was driven by selection of substitutions that reduce the rate of superoxide production thereby increasing songbird lifespan and cognitive abilities.

KEY WORDS: Songbirds, Cytochrome b, Bc1 complex, ROS, Superoxide

## INTRODUCTION

The basal rate of mitochondrial reactive oxygen species (ROS) generation varies greatly between organisms but is generally predicted to correlate with their mass-specific basal metabolic rate. Indeed, it was suggested previously that longevity of all animals is determined by their metabolic rates, the so-called 'rate of living' theory (Speakman, 2005). In both birds and mammals, basal mass-specific metabolic rates are negatively correlated with body mass, while longevity is positively correlated with body mass (Speakman, 2005) most likely reflecting the effect of metabolic

New Hope Biomedical R&D, 23 W Bridge Street, New Hope, PA 18938, USA.

*Author for correspondence (rotteh@hotmail.com)

 H.R., 0000-0003-1353-9016

rates on ROS generation. However, it is apparent that there are animal species that generate much less ROS than predicted based on their size or metabolic rates (cf. Munro et al., 2013). Both in mammals and birds there are clades that exhibit exceptions to these general rules. For instance, in mammals, anthropoid primates exhibit exceptional longevity, despite exceptionally high metabolic rates, while also exhibiting accelerated evolution of the mitochondria complexes that generate ROS, including cytochrome b (Rottenberg, 2007a; Rottenberg, 2014). Similarly, in birds, songbirds exhibit exceptional longevity, despite exceptionally high metabolic rates, that is also correlated with high rate of cytochrome b evolution (Rottenberg, 2007b). We hypothesized that in these clades the substitutions observed in cytochrome b (and other mtDNA coded peptides) were positively selected to reduce the production of ROS and that such adaptive selection will be manifested by accelerated evolution of mtDNA. It has been demonstrated that both in mammals (Lambert et al., 2007) and birds (Delhaye et al., 2016) longevity is negatively correlated with the production of ROS by mitochondria.

The bc1 complex is a critical component of the oxidative phosphorylation system (OXPHOS) that catalyzes the oxidation of a variety of substrates while utilizing the released energy for an efficient synthesis of ATP. In eukaryotes, OXPHOS is located in the inner mitochondrial membrane and is composed of supercomplexes of the electron transport chain: complexes I, II, III and IV in various combinations – and of ATP synthase (complex V). Three of the electron transport complexes: complex I (NADH dehydrogenase), complex III (bc1, ubiquinol-cytochrome c oxidoreductase), and complex IV (cytochrome c oxidase) couple their redox reactions to pump protons across the inner membrane thereby generating a proton electrochemical potential gradient (protonmotive force), which drive the synthesis of ATP by reversing the flow of proton throw the forth proton pump, complex V, i.e. ATP synthase (Mitchell, 1979).

The redox reactions in complex I and III and to some extent also complex II are associated with an electron leak due to the interaction of oxygen with their free radical intermediates leading to the release of superoxide that leads to the formation of other reactive oxygen and nitrogen species, ROS (Wong et al., 2017).

The release of ROS from the bc1 complex, which is very sensitive to the metabolic state of the mitochondria, serves as a signal from the mitochondria to many cellular processes in the cytoplasm, the nucleus and beyond (cf. Weinberg et al., 2019; Cabello-Rivera et al., 2022; Homma et al., 2021). However, excess production of mitochondrial ROS, that result from mitochondrial dysfunction, and is pronounced in aging, is very destructive to cellular proteins, phospholipids, and particularly DNA, and lead to degenerative diseases, neurodegeneration and eventually cell death (Dai et al., 2014; Rottenberg and Hoek, 2017; Rottenberg and Hoek, 2021).

The catalytic form of the bc1 complex is a dimer. At the core of each monomer of the bc1 complex there are three subunits: cytochrome b, cytochrome c1 and the Rieske [2Fe-2S] protein. These three proteins are necessary and sufficient to catalyze the

oxidation of ubiquinol by the reduction of cytochrome c. Cytochrome b is composed of eight transmembrane helices containing two b-type hemes, a low potential heme B**L** close to the intermembrane space, the P-side, and a high potential heme B**H** on the other side of the protein close to the matrix space, the N-side (Fig. 1). Adjacent to heme B**L** is a binding site for ubiquinol (Qo), while adjacent to heme B**H** is a binding site for ubiquinone (Qi). Cytochrome c1 has only one transmembrane helix while its head, that contains heme c, is located on the P-side. Similarly, the Rieske protein has only one transmembrane helix but the head of the protein, that contains the [2Fe-2S] redox center is also on the P-surface of the cytochrome b of the adjacent bc1 monomer, able to reach its Qo site and also move after reduction to heme c of the adject monomer (Fig. 1). The mitochondrial bc1 complex contains eight additional proteins that are not necessary for its catalytic activity with largely unknown functions (Xia et al., 2013). However, one of these subunits, the 14KDA peptide, which is located on the N-surface just above the Qi site and appears to interact with critical residues of cytochrome b, was also included in our study (Fig. 1).

The mechanism by which bc1 catalyzes the oxidation of ubiquinone by cytochrome c while generating a protonmotive force, the modified Q cycle, is now well established (Mitchell, 1975; Crofts, 2021). In short, as outlined in Fig. 2, the cycle starts by ubiquinol binding to the Qo site and the transfer one electron to the

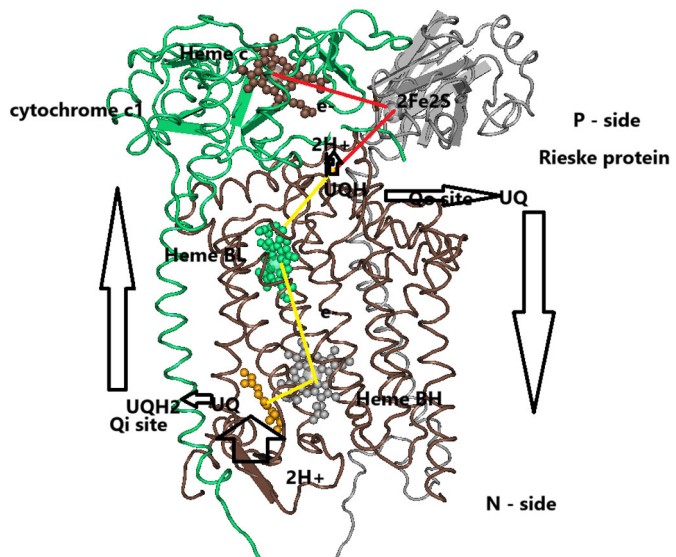

**Fig. 2. The Q cycle.** The cycle starts by binding of ubiquinol (QH2) at the Qo site on the P-side of cytochrome b (top, right). The [2Fe-2S] redox center of the Rieske protein binds to Ubiquinol and accept one electron and a proton and the reduced headgroup move along the p-surface towards heme c of cytochrome c1, release the proton and deliver the electron to heme c, which delivers the electron to cytochrome c. This is the high-potential electron pathway (red). Simultaneously the second ubiquinol electron is delivered to the low-potential electron pathway (yellow): heme B**L** (green) to heme B**H** (gray) to ubiquinone (orange) at the Qi site. The anion semiquinone then binds a proton from the N-side to form the neutral semiquinone QH, thus completing half the cycle. The oxidized ubiquinone at the Qo site is released and a second ubiquinol binds at the Qo site and the oxidation of ubiquinol is repeated, while the second electron arriving at the Qi site fully reduces the semiquinone that accepts a second proton, thus forming a ubiquinol that is released and can diffuse to the P-side, completing the Q cycle. Adapted from PBD: 3H1H.

[2Fe-2S] redox center of the Rieske protein which is also bound at the Qo site; this state is called the 'b' state of the complex. After the reduction of the [2Fe-2S] center and the release of a proton to the P-surface the whole Rieske head group move along the P-surface toward the cytochrome c1 head and transfer the electron the heme c; this state of the bc1 is called the 'c' state (Fig. 2). Simultaneously, the second ubiquinol electron is transferred to the low potential heme B**L** and the second proton is released to the P side. Heme B**L** transfers its electron across the membrane to the high potential heme B**H**, thereby generating a membrane potential. Then, heme B**H** transfers the electron to a ubiquinone bound at the Qi site, while ubiquinone accepts a proton form the N-side thus forming a semiquinone at the Qi site and further increasing the protonmotive force (Robertson and Dutton, 1988). This whole process is repeated with the oxidation of another ubiquinol at the Qo site and when the second electron arrives at the semiquinone at the Qi site it accepts another proton from the N-side thus forming a ubiquinol. This ubiquinol then diffuses to the P side thus completing the Q-cycle (Fig. 2).

It is well established that during the Q-cycle superoxide is generated by oxygen interaction with the semiquinone that is transiently formed at the Qo site (Dröse and Brandt, 2008; Quinlan et al., 2011). Inhibition of electron flow from B**L** to B**H** (and therefore increase backflow of electron from B**L** to ubiquinone) would increase the generation of superoxide at the Qo site (Pagacz et al., 2024). Antimycin A that binds to the Qi site and inhibits electron flow from B**L** to B**H,** or high membrane potential that inhibit ubiquinone reduction increase the rate of superoxide generation (Rottenberg et al.,

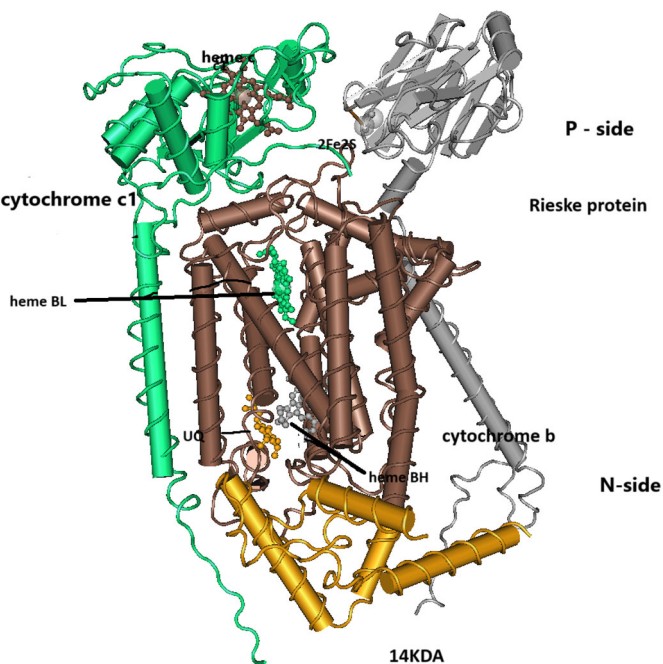

**Fig. 1. The catalytic core of the bc1 complex.** The catalytic form of bc1 is a dimer (bc1)$_2$ composed of two identical monomers of the bc1 complex. At the center of the catalytic core of each monomer is cytochrome b (brown) composed of eight transmembrane helices that contain heme B**H** (green) near the N-side (matrix), and heme B**L** (grey) near the P-side (intramembrane space). Ubiquinone (orange) is bound at the Qi site adjacent to heme B**H**. Cytochrome c1 (green) has one transmembrane helix adjacent to cytochrome b and a large head group that contains heme c (brown). The Rieske protein (grey) single transmembrane helix is bound to the second bc1 monomer (not shown) but its large headgroup on the P-side, that contain the [2Fe-2S] center (grey), can reach both the ubiquinol binding site Qo adject to heme B**L** and move closer to heme c of cytochrome c1 when in the c state as shown here. The 14KDA subunit (orange) while not generally considered part of the catalytic core is on the N-side close to the Qi site interacting with residues of this site. Adapted from PBD:3H1H.

2009). In general, the lifetime of the semiquinone in the Qo site will determine the rate of superoxide generation.

We have recently shown there are human (and other anthropoids) specific substitutions in bc1 peptides that are predicted to modulate the rates of both ubiquinol oxidation at the Qo site and ubiquinone reduction at the Qi site (Rottenberg, 2023). These findings are compatible with the hypothesis that in human (and other anthropoids) the bc1complex evolved to reduce the production of superoxide (Rottenberg, 2014). We have previously shown that like anthropoid primates, songbirds exhibit exceptional longevity, exceptionally high metabolic rates, and accelerated evolution of cytochrome b when compared to other modern birds (Rottenberg, 2007b). Here, we analyze the predicted effects of the songbirds-specific cytochrome b substitutions on bc1 function and conclude that most of the songbird-specific substitutions in cytochrome b were selected to modulate the rates of ubiquinol oxidation at the Qo site or ubiquinone reduction at the Qi site, and therefore, most likely, the rate of superoxide generation.

## RESULTS

### Mitochondria ROS generation is much lower in songbirds than that of birds from other orders

Delhaye et al. (2016) showed that mitochondrial ROS production is correlated with longevity in birds. Delhaye et al. (2016) measured ROS production in birds from seven different orders including nine songbirds species but did not compare mitochondria ROS production in songbirds to that of bird species from other orders. Because mitochondrial ROS production depends on the rate of respiration, we normalize their values of mitochondrial ROS (Table 1, Delhaye et al., 2016) by dividing with each species mass-specific basal metabolic rates (also calculated from the results of their Table 1). Separating the nine songbird species from all other 11 bird species that represent six other modern bird orders, we find that the normalized mitochondrial ROS production in songbirds is much lower than that of all other birds 409(au) compared to 1449(au), a 72% reduction in normalized mitochondrial ROS production ($P<0.0129$). Moreover, only in songbirds, exceptional longevity (i.e. longevity/body mass) was very strongly and negatively correlated with the normalized mitochondria ROS production, R=−0.702 compared with R=−0.394 for all other modern birds (Dataset S1 for the detailed calculations).

### The accelerated evolution of bc1 subunits in songbirds

In the alignment of 59 cytochrome b sequences from all birds' orders assembled for this study (Dataset S3) the average genetic distance of songbirds (Passeri) species from a Palaeognathae species (Apteryx) is 0.198±0.013 while the average distance from

Apteryx of species from all other modern birds orders is only 0.131±0.019, a 51% increased rate of substitutions in songbirds ($P<0.001$). Similarly, the evolution of the other catalytic subunits, cytochrome c1 and the Rieske [2Fe-2S] protein, is also accelerated in songbirds: for cytochrome c1: 0.137±0.014 versus 0.093±0.022 in modern birds ($P<0.002$), a 47% acceleration, and 0.137±0.010 versus 0.074±0.018 in modern birds, a 85% acceleration ($P<0.001$) for the Rieske protein. The evolution of the nuclear DNA-coded 14KDA subunit is also greatly accelerated in songbirds: 0.166±0.022 versus 0.114±0.030 in modern birds, a 46% acceleration ($P<0.001$). These findings strongly suggest that the bc1 complex in songbirds evolved to modulate the catalytic activity of the bc1 complex.

### Songbirds' specific substitution in the bc1 complex

There are 21 significant songbird-specific substitutions in cytochrome b (Dataset S3). We consider a substitution as significant only when either the charge, hydrogen bonding capacity, aromaticity or hydrophobicity are modified by the substitution. Seventeen significant songbird-specific substitution are observed in all songbirds, whereas four substitution – S26P, M54T, L210T and T328A – are only observed in the superfamily Fringillidae. Most of the songbird-specific substitutions, twelve, are on the N-side (matrix) surface of the bc1 complex while five substitutions are on the P-side surface (intermembrane space), and four substitutions are located within transmembrane helices.

Eight substitutions, six on the N-side surface, and two on the transmembrane helices, are located within 10 Å of either heme B**H** or the adjacent ubiquinone binding site Qi or both (Fig. 3 and Table 1). The redox potentials of protein-bound hemes and quinones are strongly dependent on the local dielectric properties of the surroundings residues (Gibney et al., 2001; Olson et al., 2013) and therefore these eight substitutions are likely to have some effect on the redox potentials of heme B**H** and the ubiquinone bound at the Qi site. Moreover, all these substitutions, except A191V, interact directly with either heme B**H**, ubiquinone or residue in the ubiquinone site Qi, as described in Table 1 and below.

### Substitutions of residues that interact directly with heme bH and the Qi site

Heme B**H** is located near the matrix surface (N-side) of cytochrome b (Fig. 1). In the presence of high protonmotive force (high membrane potential and high ΔpH), as occur in vivo in the resting state (mitochondrial state 4), the Q-cycle is slowed down and under these conditions the rate of the Q-cycle depends on the rate of the reduction of ubiquinone at the Qi site and therefore the redox potentials of heme B**H** and the Qi-bound ubiquinone. These are also

**Table 1. Songbirds-specific bc1 substitution that are located close to heme BH and/or the Qi quinone binding site**

| Substitution | Distance from heme B**H** | Distance from ubiquinone | Distance from antimycin | Interactions |
|---|---|---|---|---|
| S18A | <8.5 Å | <4.0 Å | <4.0 Å | Contacts: AA, UQ |
| S26P | <10.0 Å | <9.0 Å | <9.0 Å | HB: D217(b), N27(b) Contact: L70(f), W227(c) |
| A30T | <7.5 Å | <9.0 Å | <8.0 Å | HB: CDL1 Contact CDL1, 2 |
| Y110N | <4.5 Å | <8.0 Å | <8.5 Å | π-cation and HB: R314(b) Contact: heme B**H** |
| T116I,V | <6.0 Å | >10.0 Å | >10.0 Å | HB: H197(b) |
| A191V | <6.5 Å | <10.0 Å | <10.0 Å | |
| L210T | <10.0 Å | >10.0 Å | >10.0 Å | Contact: A65(f), L66(f), S69(f) |
| F225Y | <8.5 Å | <4.5 Å | <4.5 Å | π-cation and HB: R314(b)-Y225(b) Contact: heme B**H** |

UQ, ubiquinone; AA, antimycin A; CDL, cardiolipin; HB, hydrogen bond; bc1 subunits labels: cytochrome b, (b); cytochrome c1, (c); Rieske protein, (r); 14KDA, (f).

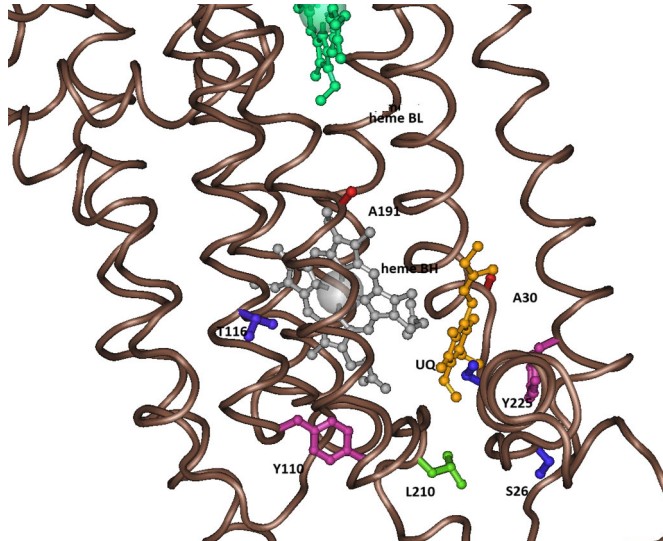

**Fig. 3. Songbird-specific substitutions in the proximity of heme BH and ubiquinone at the Qi site.** The substituted modern-birds residues are (clockwise from top): alanine A191V (red), alanine A30T (red), serine S18A (blue), tyrosine Y225F (crimson), serine S26P (blue), leucine L210T (blue), tyrosine Y110N (crimson), threonine T116I, V (violet). See Table 1 and text for more details. Adapted from pdb:3H1H.

the conditions that are known to accelerate the production of superoxide (Rottenberg et al., 2009). In songbirds, there are two cytochrome b substitutions of residues that interact with heme BH - T116I,V(b) and Y110N(b) (Fig. 4), and two substitution in the Qi site -S18A(b) and F225Y(b) (Fig. 5), and these

substitutions may modulate the rate of the reduction of ubiquinone at the Qi site.

The most important substitution is most likely the threonine substitution T116I,V(b). T116(b) forms a very strong hydrogen bond (2.7 Å) with histidine H197(b), one of the two histidine residues [H197(b), H98(b)] that ligate the iron of heme $B_H$ in cytochrome b. Since the redox potential of cytochrome b hemes strongly depends on the ligating histidine (Pintscher et al., 2016) a hydrogen bond to the second histidine nitrogen (ND1) of H197(b) should affect $B_H$ redox potential. In all birds, both in heme $B_L$ and heme $B_H$, one ligating histidine is hydrogen bonded with threonine. The threonine that is hydrogen bonded to histidine H84(b) of heme $B_L$, T48(b), is strictly conserved in all birds but the threonine, T116(b), that is hydrogen bonded to histidine H197(b) of heme $B_H$ is conserved in all orders of birds except songbirds in which it is substituted with a hydrophobic residue that cannot form a hydrogen bond, either isoleucine or valine. Therefore, it can be predicted that this substitution modulates the redox potential of heme $B_H$.

Another substitution that may directly affect the redox potential of heme $B_H$ is the tyrosine substitution Y110N(b). This tyrosine forms a weak hydrogen bond (3.5–3.9 Å) and a strong $\pi$-cation bond with arginine R314(b) (3.6 Å). Arginine R314(b) in turn form a salt bridge with the D-propionate of heme $B_H$ (a range of 5.3-6.2 Å in different structures of cytochrome b). This salt bridge affects the redox potential of heme $B_H$ because it reduces the electrostatic effect of the D-propionate carboxylate on the heme $B_H$ potential (Das and Medhi, 1998). In addition to the $\pi$-cation bond between R314(b) and Y110(b) there is another $\pi$-cation bond that holds R314(b) in its position: that of R314(b) and Y38(f) of the 14KDA subunit. In the 14KDA subunit there are also songbird-specific substitutions (Dataset S6) one of which is the substitution Y38N(f).

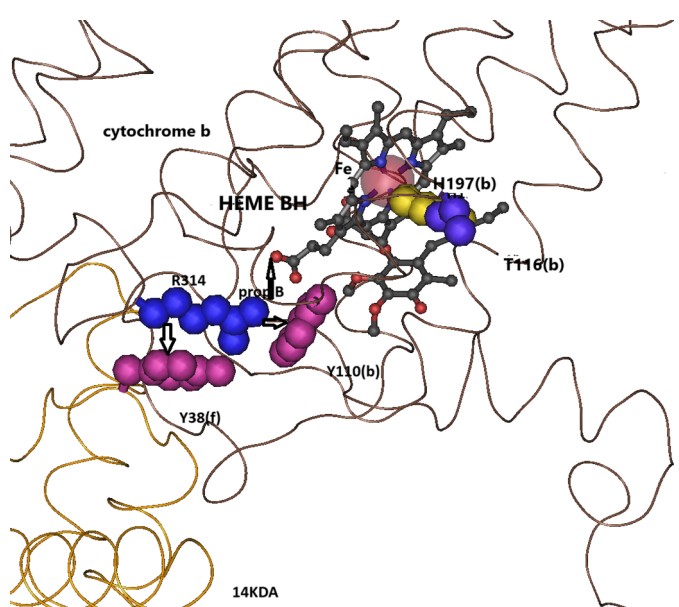

**Fig. 4. Songbird-specific substitutions of residues that interact directly with heme BH.** The strong hydrogen-bond between threonine T116(b) and histidine H197(b) that ligates the iron of heme $B_H$ is substituted with isoleucine or valine, which eliminates this hydrogen bond. The $\pi$-cation bonds tyrosine Y110(b) with arginine R314(b), and tyrosine Y38(f) (of the 14KDA subunit) with R314(b) are eliminated by the substitutions Y110N(b) and Y38N(f), which will modulate the salt-bridge between R314(b) and the B-propionate of heme $B_H$. Adapted from PDB:3H1H.

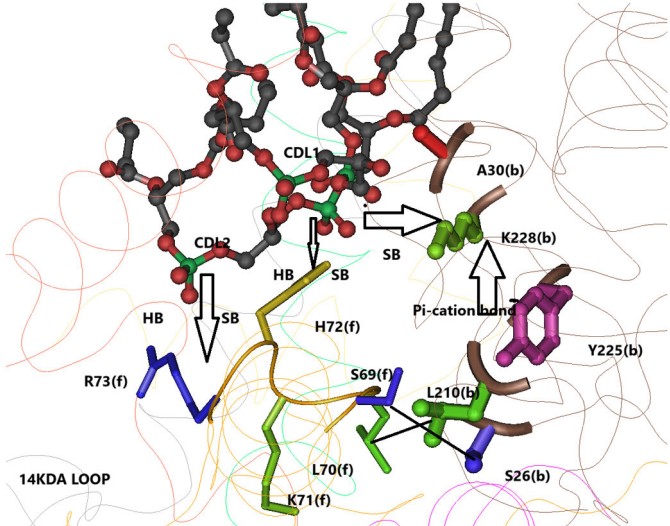

**Fig. 5. Songbird-specific substitutions that may affect the pKa of lysine K228(b).** Alanine A30 interacts with the head-group of a cardiolipin that forms a strong salt-bridge with K228(b); the substitution A30T may affect this salt-bridge, thus modulating the pKA of K228(b). Tyrosine Y225(b) forms a $\pi$-cation bond with K228(b); in most modern birds there is a phenylalanine in this position F225(b); the predicted F225(b)- K228(b) bond, if it exists, is most likely different from the Y225(b)-K228(b) bond. The substitutions S26P(b) and L210T(b) will disrupt the interactions between cytochrome b and the 14KDA subunit [serine 26(b)-L70(f) and L210(b)-S69(f)], which most likely affects the strong interactions of the 14KDA residues H72(f) and R73(f) with the cardiolipins that form salt bridges with K228(b), thus modulating the pKa of K228(b). Adapted from PDB:3H1H.

It is therefore clear that in songbirds there was a selection pressure to modulate the R314(b)-D-propionate salt bridge since the hydrogen bond [Y110(b)-R314(b)] and the two π-cation bonds with tyrosine Y110(b) and Y38(f) that keep R314(b) sidechain extended toward the D-propionate of heme B**H** are both eliminated by substitutions in songbirds (Fig. 4).

Serine residue S18(b) is conserved in all birds except songbirds in which the substitution S18A(b) is observed. This residue is located within the Qi ubiquinone binding site. In bc1 crystal structures that contain ubiquinone in this site the serine S18(b) is in close contact with ubiquinone (3.6 Å). While it is not clear whether serine S18(b) forms a hydrogen bond with ubiquinone at some point during ubiquinone reduction, its replacement with alanine should affect the redox potential of ubiquinone at the Qi site.

### Substitutions that may affect the pKa of lysine K228(b) in the Qi site

It is believed that lysine K228(b), which is located within the Qi site, participates in proton transfer to ubiquinone during its reduction by heme B**H** (Kuleta et al., 2016; Postila et al., 2016). Therefore, any modulation of the interactions of this lysine with other residues or cardiolipins may affect this reaction.

Chicken, like songbirds, also exhibits the substitution F225Y(b). Tyrosine Y225(b) is located within the Qi site. It forms close contact with antimycin and is <4.5 Å from ubiquinone. Most importantly, it forms a π-cation bond with lysine K228(b) within the Qi site (5.9 Å in PDB:3H1I) (Fig. 5). In the mammalian bc1 there is also tyrosine in this position – Y224(b), which forms a stronger π-cation bond with K227(b) (5.0 Å in PDB:1NTZ). The phenylalanine residue F225(b) is conserved in all birds' orders except songbirds; however, the substitution F225Y is also observed in few species like chicken and falcon (supplementary material, cytochrome b alignment). While it is quite possible that F225 may also form a π-cation bond with K228(b), the latter bond, if it exists, most likely would be somewhat different from the Y225(b)-K228(b) bond that is observed in songbirds. Therefore, this songbird-specific substitution most likely modulates the pKa of K228(b).

Lysine K228(b) forms several salt bridges with two cardiolipins, 2.9 Å-4.9 Å in different chicken bc1 structures (Fig. 5). These salt bridges should have strong effects on the pKa of lysine 228(b). Indeed, cardiolipins appear to be necessary for bc1 function (Lange et al., 2001; Hielscher et al., 2009; Wenz et al., 2009; Postila et al., 2016). In the chicken structure these cardiolipins are anchored by several strong hydrogen bonds (3.5 Å) and strong salt bridges (2.6–4.0 Å) with two positively charged residues of the 14KDA subunit-histidine H72(f) and arginine R73(f) (Fig. 5). In most bird species, including songbirds, R73(f) is substituted with glutamine Q73(f). R73(f) is only observed in chicken and few other related bird species. In mammals, Q72(f) is observed in the bovine structure in this position, where it forms hydrogen bonds with the cardiolipin that interact with K227(b). Therefore, it is very likely that in birds, as well, Q73(f) forms hydrogen bonds with a cardiolipin that interact with K228(b).

There are two songbird-specific substitutions in cytochrome b, S26P(b) and L210P(b) (Fig. 5), which may modulate the interactions between the 14KDA residues H72(f), Q73(f) and the cardiolipins, and therefore also modulate the salt bridges between the cardiolipin and K228(b).

Serine S26(b) is in contact with L70(f) close to the 14KDA cardiolipins' binding sites. In addition, S26(b) forms a strong hydrogen bond with aspartate D217(b), which form salt bridges with 14 KDA residue K63(f). The substitution S26P will eliminate

the hydrogen bond with D217(b) and modulate the interactions with K63(f) and L70(f) of the 14KDA subunit, most likely affecting the hydrogen bonds and salt-bridges between the 14KDA subunit and cardiolipins.

Leucine L210(b) is in contact with S69(f). S69(f), which is in contact with a cardiolipin, is hydrogen bonded with N27(b). The hydrogen bond N27(b)-S69(f) is very strong (3.0-3.3 Å) in the c-like structures and broken in the b-like structures (3.5-3.9 Å). The strength of this bond is negatively correlated with the strength of the salt bridges of the cardiolipins with K228(b) (2.9-4.9 Å). Therefore, the substitution L210 T(b) most likely modulates the N27(b)-S69(p) bond and affects the cardiolipins interaction with K228(b).

Another cytochrome b substitution in songbirds, A30T(b), may affect the interactions of cardiolipin with K228(b). Alanine A30(b) is in very close contact with a cardiolipin headgroup that forms a salt-bridge with K228(b), and its backbone nitrogen forms a strong (3.0 Å) hydrogen bond with that headgroup (Fig. 5). The substitution A30T introduces a much larger threonine headgroup that in all likelihood will disrupt this interaction and modulate the salt-bridge between the cardiolipin and K228(b). Thus, it appears that the cluster of cytochrome b substitutions – S26P(b), A30T(b), and L210L(b) were selected to modulate the salt bridges between cardiolipin 1 and 2 and K228(b), thus modulation the rate of protonation of ubiquinone at site Qi.

### Additional substitutions in songbirds' cytochrome b that affect the interactions between cytochrome b and 14KDA subunit

Three more songbird-specific cytochrome b substitutions on the matrix surface of cytochrome b, Q313L(b) and the carboxy terminal residues N379K(b) and Y380L(b), interact with the 14KDA subunit. Glutamine Q313(b) forms a very strong hydrogen bond with T36(f), which will be eliminated by substitution Q313L. The cytochrome b carboxyterminal substitutions N379K(b) and Y380L(b) also modulate the interaction between cytochrome b and the 14KDA protein. N379(b) forms hydrogen bonds with R33(f) and is also in contact with R17(f) and E91(f) and the substitution N379K(b) may weaken the contact with R33(f) but increase the contact (perhaps forming a salt bridge) with E91(f). In songbirds there is also a 14KDA substitution R17C(f), which further supports the notion that these substitutions were selected to modulate the contact between the carboxyterminal of cytochrome b and 14KDA. The tyrosine Y380(b) forms a hydrogen bond and a π-cation bond with R33(f) and also a hydrogen bond with D34(f). The substitution Y380L(b) will eliminate these bonds. It is therefore apparent that these substitutions, N379K(b) and Y380L(b), were selected to weaken the bond of cytochrome b with the 14KDA residues R33(f), D34(f) and possibly increase the bond with E91(f). It is not possible to predict what effect these structural modifications will have on the function of the bc1 complex.

### Substitutions of cytochrome b 'vise' residues that interact with the 'hinge' region of the Rieske protein

As described above, the oxidation of ubiquinol at the cytochrome b Qo site by the [2Fe-2S] center of the Rieske protein is followed by the large scale movement of the Rieske protein headgroup from the Qo site on cytochrome b surface toward the heme c of cytochrome c1, allowing the transfer of electron from the [2Fe-2S] group to heme c (Berry et al., 2013; Xia et al., 2013). This movement is enabled by the conformation change of the 'hinge' region of the Rieske protein through its interactions with the 'vise' region which

is provided by the two adjacent cytochrome b peptides and cytochrome c1 residues. Most of the residues of the two 'vise' region in cytochrome b and the 'hinge' region of the Rieske protein are highly conserved in all forms of the bc1 complex. In songbirds there are three substitutions in the 'vise' region of one monomer of the cytochrome b (**L**51-**I**80): M54T(b), N73D(b) and Y76F(b) (Fig. 6). Interestingly, in songbirds there are no substitutions in the 'vise' region of the second monomer of cytochrome b (**E**163-**R**178); whereas, in anthropoid primates that also exhibit substitution in the 'vise' region of cytochrome b, all the substitutions are in the first monomer alone (Rottenberg, 2023). The most significant substitution appears to be N73D(b). This substitution is a reversal to the commonly observed D72(b) in most mitochondrial cytochrome b other than that of birds. The asparagine N73(b) interacts strongly with several residues of the Rieske hinge region, particularly when the bc1 is in the c conformation (e.g. PDB:3H1H). There are 20 close (<4 Å) hydrophobic contacts with S65(r), A66(r), and D67(r). In addition, there is a strong hydrogen bond (2.8 Å) between backbone atoms of N73(b) and A66(r) and a weaker hydrogen bond (3.5 Å) between the head groups of N73(b) and D67(r). When the bc1 complex is in the b conformation (e.g. 3H1I) these N73(b) contacts are considerably weaker: while the backbone hydrogen bond between N73(b) and A66(r) is conserved the hydrogen bond between N73(b) and D67(r) is broken and the number and strength of hydrophobic contacts between N73(b) and the Rieske 'hinge' is greatly reduced. It can be predicted that in songbirds the contact between D73(b) and Rieske protein, particularly in the c conformation, will be weaker than that of N73(b) in other birds because of the expected electrostatic repulsion between D73(b) and D67(r). These predictions can be tested by examination of the bc1 complex of mammals (e.g. bovine) where the corresponding residue is D72(b): in the bovine structures (e.g. PDB:1PPJ) there is no hydrogen bond between D72(b) and D67(R) while the interaction between D72(b) and R49(c) and H70(c) become very strong through the formation of hydrogen bonds and salt bridges.

The substitution M54T(b) should also influence the Riske hinge interactions with cytochrome b. M54(b) is in strong contact (3.2 Å) with S61(r) and a weaker contact (3.6 Å) with L62(r) in addition to contact with residues on the adjacent cytochrome b. The substitution Y76F(b) could also modify the interactions of tyrosine Y76(b) with the 'hinge' region of the Rieske protein. This residue is in contact with both Q57(r) and S60(r). In addition, Y36(b) hydroxyl group appears to be a ligand binding site, binding glycerol in one structure

and phosphatidyl choline in another. Therefore, this substitution may modify the interactions with the Riske 'hinge' region and also eliminate a phospholipid binding site.

Because most of the cytochrome b 'vise' residues are highly conserved in all forms of bc1 It can be predicted that these three substitutions will most likely inhibit the movement of the [2F-2S] headgroup and therefore may inhibit the rate of superoxide production (see Discussion).

The functional modulations that can be predicted by the examination of the songbird-specific substitutions in cytochrome b and 14KDA reported here are grouped in four clusters: two clusters of substitutions that are predicted to modulate the redox potentials of heme B**H** and ubiquinone at the Qi site, a cluster of substitutions that can be predicted to modulate the rate of the protonation of ubiquinone at the Qi site by lysine K228(b) and thus modulate the rate of reduction of ubiquinone at the Qi site, and a cluster of cytochrome b substitutions that can be predicted to affect the interactions of the Rieske protein 'hinge' domain with the cytochrome b 'vise' domain and thus modulate the rate of oxidation of ubiquinol at the Qo site.

## DISCUSSION

In songbirds the rate of cytochrome b evolution correlated with exceptional longevity (Rottenberg, 2007b). Because longevity is negatively correlated with the rate of ROS generation (cf. Lambert et al., 2007) we suggested that, in songbirds, cytochrome b evolved to reduce the production of superoxide. This suggestion is supported by our analysis of the data from the study by Delhaye et al. (2016) (supplementary information, mitochondrial ROS Delhaye), which shows that normalized mitochondrial ROS production is much lower in songbirds compared to birds from other orders. Moreover, exceptional longevity was much more strongly correlated with normalized mitochondrial ROS production in songbirds than in other birds. This suggests that in songbirds the difference in exceptional longevity between species largely depends on mitochondrial ROS production, and to a lesser extent on other factors. Since we previously reported that exceptional longevity correlated strongly with the rate of evolution of cytochrome b only in songbirds (Rottenberg, 2007b), these data together support the hypothesis that cytochrome b evolution in songbirds was driven by adaptive evolution for reduced production of superoxide.

The redox potential of protein-bound hemes and quinones is greatly affected by the dielectric environment in which they are embedded; the hydrophobicity of nearby residues, their charge, aromaticity and the presence of water molecules (Gibney et al., 2001). Moreover, direct interactions with the protein bound heme or quinones – covalent bonds, hydrogen bond, salt bridges, π-cation bonds, etc., greatly affect the redox potential of these factors (Olson et al., 2013;Kuleta et al., 2021). The redox potential of both heme B**H** and ubiquinone bound in the Qi site should be affected to some degree by the eight substitutions of residues that constitute the dielectric environment of these ligands because all of these substitutions change the nature of the residue (Table 1, Fig. 3). In addition, several of these substitutions directly affect residues that interact directly with heme B**H**: the threonine substitution T116V, I(b) eliminates a hydrogen bond with histidine H197(b) that ligates Fe in heme B**H**. The cytochrome b tyrosine substitutions Y110N(b), as well as the 14KDA tyrosine substitution Y38N(f) eliminate the π-cation bonds with arginine R313(b) that form a salt bridge with propionate B of heme B**H**. Because the heme propionates negative charges decrease the redox potential of heme B**H**, the salt bridge

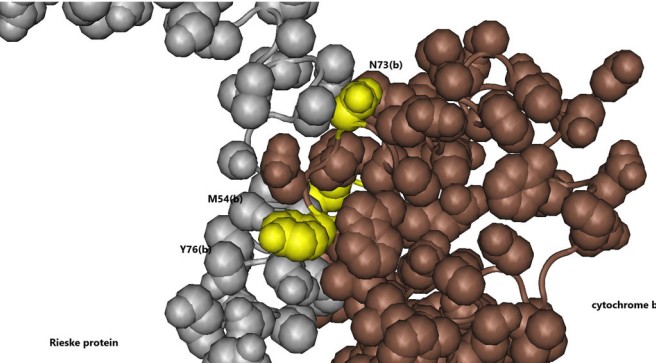

**Fig. 6. Songbirds-specific substitutions that affect the interactions between the 'vise' cytochrome b region and the 'hinge' Rieske protein region.** The cytochrome b substitutions on one face of the 'vise' region - M54T(b), N73D(b) and Y76F(b). Adapted from PDB:3H1I.

with arginine increase the redox potential. It has been argued that the redox potentials of cytochrome b hemes have little effect on either the turnover rate of the Q cycle (Pintscher et al., 2016) or the rate of production of superoxide (Pagacz et al., 2024). However, these conclusions are based largely on experiments with isolated bacterial complexes, where the turnover rate depends on the rate of oxidation of ubiquinol at the Qo site, which does not depend on the cytochrome b hemes. *In vivo*, particularly at the resting metabolic state (mitochondrial 'state 4'), the turnover of the Q cycle becomes much slower (respiratory control) and depends on the rate of reduction of ubiquinone at the Qi site, which in turn depends on the magnitude of the protonmotive force. Under these conditions redox potentials of heme B**H** and ubiquinone at the Qi site do have a larger effect on the rate of the Q cycle turnover and expected to have even stronger effect on the rate of superoxide reduction (see below).

A novel finding of this study is that there are two clusters of songbirds-specific cytochrome b substitutions of residues that interact with the 14KDA subunit. There is little experimental evidence for a functional role of the 14KDA subunit except for one study that showed that mild proteolytic treatment of the matrix surface of the bc1 complex, which removed a hydrophilic segment of the 14KDA subunit, decreased the H+/e- ratio of the bc1 complex (Cocco et al., 1991). Two of the songbird-specific substitutions, S26P(b) and L210T(b), modulate cytochrome b interaction with 14KDA residues -S26(b)-L70(f) and L210(b)-S69(f). These 14KDA residues are on a loop close to the cationic residues H72(f), R73(f), which strongly bind two cardiolipins that form strong salt bridges with lysine K228(b). We suggest that these substitutions were selected to modulate the salt bridges between the cardiolipins and K228(b), and, thus modulate the protonation of ubiquinone during the reduction of ubiquinone in site Qi. Further evidence that these substitutions were selected to modulate the interaction of cardiolipin with K228(b) is the fact that S69(f) interacts directly with a cardiolipin that interacts with K228(b). Moreover, another songbird-specific substitution is alanine A30T(b), where A30(b) interacts strongly with a cardiolipin headgroup that forms a salt bridge with K228(b). Furthermore, the fact that the strength of the hydrogen bond asparagine N27(b)-S69(f), which varies in different structures of bc1, appears to affect the strength of the salt bridges of cardiolipins with K228(b), suggests that modulating the interaction S26(b) with S69(f) by the substitution S26P(b) will also modulate the interaction of the 14KDA bound cardiolipins with K228(b). Finally, the songbird-specific substitution F225Y(b) that results in a π-cation bridge Y225(b)-K228(b) adds further evidence to the proposition that a cluster of songbird-specific substitutions were selected to modulate the pKa of K228(b) and thereby affected the rate of ubiquinone protonation during ubiquinone reduction in the Qi site.

There were also a few songbird-specific substitutions that cluster around the 'vise' domain of cytochrome b, which interact with the 'hinge' domain of the Rieske protein. These substitutions – M54T(b), N73D(b) and Y76F(b) – were apparently selected to modulate the movement of the Rieske head group from the b position to the c position, most likely inhibiting the rate of ubiquinol oxidation in the Qo site.

In addition to the effect of membrane potential (Rottenberg et al., 2009), the rate of superoxide generation also depends on the detailed kinetics of the Q cycle. Since the detailed kinetic constants of the many steps of the Q cycles are not known for any bc1 complexes, one can only construct computational models for these reactions (Bazil et al., 2013; Guillaud et al., 2014; Markevich and Hoek,

2015; Pagacz et al., 2024). Nevertheless, one can predict that the rate of superoxide generation by the bc1 complex will depend on the lifetime of the semiquinone at site Qo, which depends on the balance between the reactions that generate semiquinone at the Qo site and the reactions that consumed it. Because we demonstrate here that in songbirds there is a cluster of substitutions that most likely modulate the rate of ubiquinol oxidation at the Qo site, and two more clusters of substitutions that modulate the reduction of ubiquinone at the Qi site, it can be predicted that the rate of superoxide generation in songbirds would be significantly different from that of other modern birds. Our analysis of the results of the measurement of mitochondrial ROS production in birds (Delhaye et al., 2016) show that mitochondrial ROS production is greatly inhibited songbirds, and since cytochrome b is a major source of mitochondrial ROS, our results are compatible with the suggestion that the songbirds' substitutions in cytochrome b will result in the reduction of superoxide production. However, there are no experimental measurements of superoxide production by the bc1 complex of birds, let alone a comparison between superoxide production by the bc1 complex of songbirds and birds from other orders. Only experiments with isolated bc1 complexes from songbirds and other birds similar to the study of superoxide production from the reconstituted yeast complex (Rottenberg et al., 2009) could provide definite proof of our conclusions.

The parallels between the evolution of cytochrome b in songbirds and anthropoid primates are striking. Like songbirds, anthropoid primates exhibit exceptional longevity, exceptionally high metabolic rates and accelerated evolution of cytochrome b. These results led to the suggestion that in both groups the evolution of cytochrome b was driven by the pressure to reduce the production of superoxide by the bc1 complex (Rottenberg, 2007a,b, Rottenberg, 2014; Rottenberg, 2023). In both cases, as evident from the results of this study and the comparison of the structure of the human bc1 complex to that of other mammals (Rottenberg, 2023), there are group-specific substitution clusters located at the Rieske protein 'hinge' contact with the cytochrome b 'vise', which are predicted to modulate the rate of ubiquinol oxidation at the Qo site. Since the two species groups evolved independently, the clusters of cytochrome b substitutions in anthropoid primates and songbirds are located on different faces of the 'vise', on two different monomers of cytochrome b. Similarly, in both songbirds and anthropoid primates there were substitutions that modulate the salt bridge of R313(b) with propionate B of heme B**H** that affects the redox potential of heme B**H**. In songbirds, the substitutions Y110N(b) and Y38N(f) released the extended R313(b), while in anthropoid primates arginine was completely eliminated by the substitution R314N(b) (Rottenberg, 2014). The cluster of substitutions that appears to modulate the salt bridges between cardiolipins and K228(b) in songbirds does not appear in mammals; however, these salt bridges are very weak in mammals compared to birds. Interestingly in the human bc1 structure, there is a cardiolipin that forms salt bridge with histidine H201(b), that was proposed to be the second hydrogen donor to ubiquinone reduction in the Qi site (Rotsaert et al., 2008). This salt bridge is also modulated in anthropoid primates by the substitution N16H(b). In the human bc1 structure histidine 16H(b) forms a salt bridge with the cardiolipin that forms a salt bridge with histidine H201(b), thereby modulating its pKa. Thus, three functionally significant structural modulations of bc1 evolved, independently, in songbirds and anthropoid primates – modulation of the Rieske protein 'hinge'-cytochrome b 'vise' interactions that affect the rate of ubiquinol oxidation at the Qo site, modulation of the redox potential of heme B**H** that affect

the reduction of ubiquinone at the Qi site, and modulation of the salt bridges of cardiolipins with site Qi proton donors that affect rate of protonation of ubiquinone at the Qi site. Since both songbirds and anthropoid primates exhibit exceptional longevity and exceptional cognitive abilities (relative to other birds or other mammals, respectively) it is reasonable to conclude that the evolution of bc1 in both groups was driven by the pressure to reduce the production of superoxide.

## MATERIALS AND METHODS

For the alignment of the amino acids sequences of cytochrome b, cytochrome c1, Rieske protein and the 14KDA bc1 subunit of modern birds, sequences were obtained from the NCBI protein collection (www. ncbi.nlm.nih.gov/protein/). Where available, reference sequences were used. For songbirds, sequences were selected to represent as many families as available, while for other modern birds, sequences were selected to represent as many orders as available. There are 59 bird species in the cytochrome b alignment: 37 are songbirds that belong to the order Passeriformes and the other 22 birds species belong to 19 other orders of birds (Dataset S3). The sequences of each protein were aligned by ClustalX2.1 program (https://clustalx.software.informer.com/2.1/) and the genetic pairwise distances of each modern birds species from the Palaeognathae species Apteryx was calculated with the MEGA-X distance program with the JTT model with Gama parameter 2 (www. megasoftware.net). The alignments of the four proteins and the detail of the calculation is given in the supplementary information, bc1 genetic distances. For analysis of the chicken bc1 complex high resolution structures we examine all 17 bc1 chicken structures that are available in the PDB collection (www.ncbi.nlm.nih.gov/Structure/pdb/); using the Cn3D 4.3 program, which was also used to prepare the figures. All calculations were performed with Microsoft Excel programs.

### Competing interests
The author declares no competing or financial interests.

### Author contributions
Conceptualization: H.R.; Data curation: H.R.; Formal analysis: H.R.; Writing – original draft: H.R.

### Funding
 Deposited in PMC for immediate release.

### Data and resources availability
All relevant data and resources can be found within the article and its supplementary information.

### Peer review history
The peer review history is available online at https://journals.biologists.com/jcs/article-lookup/doi/10.1242/bio.061908

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
