## [Peer Review File · Biology Open]

Adaptive evolution of cytochrome b in songbirds

Hagai Rottenberg

DOI: 10.1242/bio.061908

Editor: Lewis Halsey

Review timeline

Original submission: 23 January 2025

Editorial decision: 29 January 2025

First revision received: 11 March 2025

Accepted: 12 March 2025

Original submission

First decision letter

MS ID#: bio.061908

MS TITLE: Adaptive evolution of cytochrome b in songbirds

AUTHORS: Hagai Rottenberg

I have now reached a decision on the above manuscript.

The reviewer reports are shown at the bottom of this email or can be accessed, together with a copy of this decision letter, by going to:

As you will see, the reviewers raised a number of substantial criticisms that prevent me from accepting the paper at this stage.

They suggest, however, that a revised version might prove acceptable, if you can address their concerns. If you think that you can deal satisfactorily with the criticisms on revision, I would be pleased to see a revised manuscript. We would then return it to the reviewers.

At this stage, we also ask you to ensure your manuscript complies with our formatting guidelines. Provided you are able to fully address the referees' comments, we are positive about publication of your paper (we accept over 95% of revision submissions) and therefore hope you won't mind any extra work involved in reformatting your manuscript at this point.

Please ensure that you clearly highlight all changes made in the revised manuscript. Please avoid using 'Tracked changes' in Word files as these are lost in PDF conversion.

I should be grateful if you would also provide a point-by-point response detailing how you have dealt with the points raised by the reviewers in the 'Response to Reviewers' box. Please attend to all of the reviewers' comments. If you do not agree with any of their criticisms or suggestions please explain clearly why this is so.

Reviewer 1

Comments for the author

I commend the author of the manuscript (MS), "Adaptive Evolution of Cytochrome b in Songbirds", for their time and effort. This study examines the adaptive evolution of cytochrome b in songbirds, providing compelling evidence of its accelerated evolution compared to other bird species and exploring its implications for reactive oxygen species (ROS) production. The manuscript is scientifically robust and firmly rooted in established theories of mitochondrial function and evolutionary biology.

This work has significant potential to contribute to the fields of mitochondrial physiology and evolutionary biology. However, I have a few suggestions that may enhance the manuscript's clarity, accessibility, and overall impact.

General Comments:

1. The manuscript would benefit from minor editorial corrections and improved flow to ensure better readability.
2. Some sections delve deeply into technical details, which may limit accessibility for a broader audience. Summarizing these findings in simpler terms within the introduction or discussion could enhance readability.
3. The paragraphs in the introduction lack seamless transitions and occasionally use dense jargon. Additionally, the hypothesis appears in the fourth paragraph, followed by a discussion of complex b's form and mechanism. Reorganizing these elements would improve the introduction's logical flow.
4. There are inconsistencies in terminology, such as using both "pi-cation" and "π-cation." I would recommend standardizing terms throughout the manuscript.
5. The term "pressure," used to describe evolutionary changes in complex b to reduce ROS generation, is vague. A more precise term would enhance clarity.

Specific Comments:

6. "OX PHOS" should consistently be written as "OXPHOS."
7. The manuscript references a table from Delhaye et al. (2016), but this table is not included. As it is critical to the discussion, consider summarizing the key information from the table and providing it as supplementary material, with proper attribution.
8. Including a glossary table to define technical terms would improve accessibility for a broader audience.
9. Replace "surprising finding" on page 12 with "novel finding" for a more professional tone.
10. The last sentence on page 12 ("Further evidence... salt bridge with K228(b)") contains two critical points. Splitting it into two sentences would improve clarity.
11. On page 13, revise "Since two groups..." to "Since the two species groups..." for better precision.
12. The classification of substitutions into four clusters is presented later in the discussion. Moving this classification to the findings section would improve the manuscript's flow and coherence.

*Reviewer's Responses to Questions***Experimental quality**

Does each figure have the proper controls?

Yes

Were the data analyzed using appropriate statistical tests?

Yes

Reproducibility

Were experiments performed using adequate number of biological replicates?

Yes

Does the methods section provide sufficient detail to permit reproducibility?

Yes

Completeness

Are the manuscript's conclusions supported by the data?

Yes

Scholarship

Do the authors cite and discuss the merits of data that would argue for and against their conclusion?

Yes

Does the manuscript title & abstract accurately reflect the contents of the manuscript, without hyperbole?

Yes

Reviewer 2

Comments for the author

Rottenberg investigated the predicted effects of songbird-specific cytochrome b substitutions on mitochondrial physiology and specifically their potential for modulating superoxide production. The findings showed that cytochrome b underwent accelerated evolution in songbirds and together with previous findings, suggested that the structural modifications decreased the rate of superoxide production in songbird mitochondria. The manuscript is well-written, and the findings were solid. Since I am not a structural biologist, my review will mostly focus on the evolutionary and physiological aspects of the manuscript as well as the writing. I think that the introduction would benefit from a more thorough description of the evolutionary background literature, namely metabolic scaling and the rate of living hypothesis. I also feel that the methods sections lack substantial details, but perhaps the details were included in the supplements which I do not have access to. Specific comments below.

Introduction:

The introduction section dives right into the role of bc1 complex in mitochondria bioenergetics and regulation of ROS. While this is very well-written, I think the section would benefit from a broader paragraph similar to the fourth paragraph (line 38-61) that introduces the rate of living hypothesis and how machineries for OXPHOS and ROS regulations evolve in birds and mammals.

Materials and Methods:

The author stated that sequences for modern birds are analysed. However, there is no information about which orders and how many species are analysed. I suggest either including a list of species analysed, or better a phylogeny of all the species analysed. Reading further, it looks like mammalian sequences are analysed as well, but it was not mentioned in the methods section.

Discussion:

Page 11, Line 26-33: I suggest moving this part to the methods section and adding the cited table to the manuscript either as a main table or supplement (depending on length) so that the readers do not need to look up the reference in addition.

Page 11, Line 40-44: What about other clades of birds that are either just as long-lived or longer-lived than songbirds, like many seabirds for example? Is there also evidence that there was accelerated evolution of cytochrome b and a corresponding decrease in ROS production in those groups?

Page 13-14: While the results suggested that the structural modifications could lead to decreased superoxide production, I think it is still important to include few sentences about potential caveats, especially since functional assays were not performed in the study to confirm the actual production of superoxides.

Reviewer's Responses to Questions

Experimental quality

Does each figure have the proper controls?

Yes

Were the data analyzed using appropriate statistical tests?

Yes

Reproducibility

Were experiments performed using adequate number of biological replicates?

Yes

Does the methods section provide sufficient detail to permit reproducibility?

No

Completeness

Are the manuscript's conclusions supported by the data?

Yes

Scholarship

Do the authors cite and discuss the merits of data that would argue for and against their conclusion?

Yes

Does the manuscript title & abstract accurately reflect the contents of the manuscript, without hyperbole?

Yes

First revision

Author response to reviewers' comments

I agree with most of the reviewer's comments and suggestions, and I have edited my paper and corrected my mistakes accordingly. However, it appears that the reviewers did not see my Supplementary Materials that included much of the additional information that they ask for. Also, apparently the reviewers are more interested in the topics covered in my previous publication on the subject (Rottenberg 2007 JEB) and want me to expend the discussion of this topics here, but I think that I should devote most of the introduction and discussion to the subject of this paper - the structure and function of songbirds' bc1 complex.

Reviewer 1.

General comments

1. True. I did several editorial corrections as outlined below.
2. The main topic of this paper concerns the possible effects of residue substitutions on the function and structure of bc1. One cannot avoid "delving deeply into technical details" in the results section. However, to help the reader who does not want to delve deeply into technical details I have provided four figures (Fig3-Fig6) that illustrate the most important findings in a graphic form. Moreover, these findings are described in the discussion section without the technical details.
3. Agree. I have reorganized the introduction starting with the description of the hypothesis and its origin as described in much greater detail in my previous paper on songbird's cytochrome b (Rottenberg 2007 JEB).
4. Sorry for the typographical error. it is π -cation.
5. Agree. I replaced "pressure" with adaptive evolution.

Specific comments

1. Correct. OXPPOS it is.
2. I did provide in the Supplementary Material an Excell file that contain all the relevant information from the Delhaye et al 2016 table 1, and all the calculations and statistical analysis of the results, for all the included birds, and for the separate groups of songbirds and other birds. I also moved the description of my analysis to the Result section.
3. O.K. I now provide a list of abbreviations
4. O.K.
5. O.K. I Split the sentence.
6. O.K
7. Thanks. I moved it to the end of the results.

Reviewer 2

Introduction.

The subject of this paper is the Structure and Function of songbirds' bc1 complex and therefore the bulk of the introduction must be a description of the structure and function of bc1. I have already discussed the evolutionary background for this study in great detail in my previous publication (Rottenberg 2007 JEB). I do not think it is necessary to repeat this discussion here, and I am limited to 8000 words. Nevertheless, I have moved the section that describe this background to the front of the introduction and added reference to the rate of living hypothesis (see reply to reviewer1 point 3).

Methods and Materials.

There are 59 bird species representing 20 orders in the cytochrome b alignment. I now include in the supplementary materials the list of species and their orders with their cytochrome b sequences. The alignments and its separation to songbirds and other birds orders is also included as an Excell file. The alignments of cytochrome c1, Rieske protein, and the 14KDA peptide are also included as Excell files as Supplementary Materials. I do not discuss mammalian species here. There are references to my study of human and anthropoid primates bc1 (Rottenberg 2022 BBA) and to the bovine structure of bc1 but I do not analyze mammalian sequences in this study.

Discussion.

1. Page 11 line 26-33. I did move this paragraph to the results section and the results of Delhaye et al 2016 are included in an Excell file together with all my calculations for the separate groups of songbirds and other birds (see my reply to reviewer 1 comment 7).

2. Page 11 Line 40-44. “what about other clads of birds that are either just as long-lived or longer-lived than song birds like sea birds...”. The answer to this question is to be found in my 2007 paper (JEB). In short: I define “exceptional Longevity” as distinct from “Longevity”. Songbirds, that are small and exhibit very high metabolic rates, exhibit “exceptional Longevity”, i.e. they live longer than what is expected both on the basis of their metabolic rates or body weight (i.e the deviation from the birds’ plot of Longevity ves BMRw or Longevity ves body mass are large and positive). Sea birds and other large birds exhibit high body mass and low metabolic rates and therefore do not exhibit exceptional longevity; they do not show significant deviation from the above-mentioned plots.

3. Page 13 -14. You are correct. I added this sentence to the conclusions (page 14): “However, there are no experimental measurements of superoxide production by the bc1 complex of birds, let alone a comparison between superoxide production by the bc1 complex of songbirds and birds from other orders. Only experiments with isolated bc1 complexes from songbirds and other birds similar to the study of superoxide production from the reconstituted yeast complex (Rottenberg et al, 2009) could provide definite proof of our conclusions”

Second decision letter

MS ID#: bio.061908

MS TITLE: Adaptive evolution of cytochrome b in songbirds

AUTHORS: Hagai Rottenberg

I am happy to tell you that your manuscript has been accepted for publication in Biology Open, pending our standard publication integrity checks. It was accepted on 12 Mar 2025.